# Spatial Differentiation of PM_2.5_ Concentration and Analysis of Atmospheric Health Patterns in the Xiamen-Zhangzhou-QuanZhou Urban Agglomeration

**DOI:** 10.3390/ijerph20043340

**Published:** 2023-02-14

**Authors:** Suiping Zeng, Jian Tian, Yuanzhen Song, Jian Zeng, Xiya Zhao

**Affiliations:** 1School of Architecture, Tianjin Chengjian University, Tianjin 300384, China; 2School of Architecture, Tianjin University, Tianjin 300072, China; 3School of Architecture and Urban Planning, Tongji University, Shanghai 200092, China

**Keywords:** PM_2.5_ concentration, atmospheric health pattern, positive and negative gain interval, spatial differentiation, Xiamen-Zhangzhou-Quanzhou urban agglomeration

## Abstract

Exploring the spatial differentiation of PM_2.5_ concentrations in typical urban agglomerations and analyzing their atmospheric health patterns are necessary for building high-quality urban agglomerations. Taking the Xiamen-Zhangzhou-Quanzhou urban agglomeration as an example, and based on exploratory data analysis and mathematical statistics, we explore the PM_2.5_ spatial distribution patterns and characteristics and use hierarchical analysis to construct an atmospheric health evaluation system consisting of exposure–response degree, regional vulnerability, and regional adaptation, and then identify the spatial differentiation characteristics and critical causes of the atmospheric health pattern. This study shows the following: (1) The average annual PM_2.5_ value of the area in 2020 was 19.16 μg/m^3^, which was lower than China’s mean annual quality concentration limit, and the overall performance was clean. (2) The spatial distribution patterns of the components of the atmospheric health evaluation system are different, with the overall cleanliness benefit showing a “north-central-south depression, the rest of the region is mixed,” the regional vulnerability showing a coastal to inland decay, and the regional adaptability showing a “high north, low south, high east, low west” spatial divergence pattern. (3) The high-value area of the air health pattern of the area is an “F-shaped” spatial distribution; the low-value area shows a pattern of “north-middle-south” peaks standing side by side. The assessment of health patterns in the aforementioned areas can provide theoretical references for pollution prevention and control and the construction of healthy cities.

## 1. Introduction

PM_2.5_ air pollution has a significant impact on human health risks [1,2]. Studies show [3] that even short-term exposure to air pollution at concentrations exceeding the standard can cause different degrees of coughing, breathing difficulties, and other symptoms. Long-term exposure to a heavily polluted atmosphere will induce respiratory diseases, such as chronic bronchitis and asthma, and increase the risk of cardiovascular disease and even cancer [4,5]. Therefore, it seriously endangers the public’s health. In addition, the tremendous economic losses caused by environmental pollution have a significant adverse impact on the development of the social economy. For a long time, the World Health Organization (WHO) has been concerned with instituting air quality standards. In 1987, the WHO issued the first European air quality guidelines based on health risk assessment. On 22 September 2021, the organization issued the global Air Quality Guidelines (AQG2021). The guideline values for the PM_2.5_ annual average and 24-h average were tightened to 5 μg/m^3^ and 15 μg/m^3^, respectively, a decrease of 5 and 10 μg/m^3^, respectively, compared to the 2005 version. These guidelines provide a relevant basis for establishing air quality standards in various countries. The Chinese government also updated the national standards for ambient air quality (Ambient air quality standards GB 3095–2012) in 2012. In contrast, different concentration limits are set for different types of areas, such as the annual average and 24-h average concentration limits of 15 μg/m^3^ and 35 μg/m^3^ for Class I areas, and the annual average and 24-h average concentration limits of 35 μg/m^3^ and 75 μg/m^3^ for Class II areas.

At the same time, scholars from various countries have performed extensive research on the factors affecting the spatial–temporal diffusion of air pollution and have concluded that the urban built environment has a significant impact on the mechanism of air pollution diffusion, including the urban structure, spatial form, land use, road traffic, green space, and open space [6,7,8]. For example, Wang et al. [9] took China’s prefecture-level cities as the research object and used the generalized method of moments and a dynamic panel data model to conclude that an urban spatial structure with a high population density causes air pollution to present an inverted “U” shape change and that residential land, industrial land, transportation land, and public facility land are the key factors affecting air pollution. Liu et al. [10] took Wuhan City as the research object and applied hierarchical clustering analysis to explore the relationship between air pollutants and urban form and found that the lower the building density and floor area ratio, the higher the porosity, and the lower the concentration of NO_2_ and PM_2.5_. Li et al. [11] took Maricopa County in the United States as the research object and applied regression analysis to study the impact of socioeconomic and land use factors on air pollution exposure, reaching an important conclusion that the normalized vegetation index showed a negative correlation with AOD. Du et al. [12] took Shanghai as the research object, applied the deep learning model iDeepAir and the MEIC emission inventory, and found that the traffic flow, mode, and PM_2.5_ concentration showed a positive spatial correlation.

At the same time, some scholars have also conducted relevant studies on the environmental exposure risk of pollution. For example, Xia et al. [13] applied the exposure dose assessment model, ordinary kriging space interpolation, and the Dijkstra path search algorithm to develop a smartphone search program (APP) for public health paths based on the concentration change scenario of air pollution with exposure as an indicator, to achieve the planning function of health path travel. Tong et al. [14] applied the TOA-LUR GTWR model to map the spatial and temporal distribution of PM_2.5_ in a high-density Hong Kong urban road system with high accuracy and then developed an application to provide a healthy route planning service for residents to reduce the risk of pollution exposure when they travel. Ashyeyi Mehdi et al. [15], based on the research perspective of low carbon and health, took Chicago in the United States as the research object, applied the LUR model of GBM to develop the assessment framework (iEnEx), integrating energy saving and environmental pollution exposure, and analyzed the spatial differences between household energy burden and PM_2.5_ exposure. This provides a research perspective for healthy city construction and sustainable development.

The above analytical methods for examining air pollution exposure and vulnerability are evaluation methods based on pollution exceeding thresholds and considering the degree of health effects. Additionally, they cannot evaluate the degree of health benefits for people under environmental conditions with a clean atmosphere. On the other hand, the Xiamen-Zhangzhou-Quanzhou urban agglomeration has a complex topography of interlocking mountains, cities, and sea. Comparing the Yangtze River Delta urban agglomeration and the Pearl River Delta urban agglomeration, which are also located on the eastern coast of China, we can see [16] that the spatial and temporal distribution characteristics and diffusion mechanism of PM_2.5_ concentrations in this region are characterized by the interaction between mountains, cities, and water, and the regional industrial structure is dominated by three industries and is relatively clean overall.

There is a lack of research exploring the positive and negative gain indices and atmospheric health patterns based on healthy and clean air from the perspective of people enjoying clean air. Therefore, it is necessary to explore the problem of constructing the health safety pattern of the city cluster air environment from the perspectives of environmental vulnerability, clean beneficiary surface, and risk prevention and control by combining the clustering characteristics of different functional areas and populations.

From the perspective of healthy city construction, this paper applies GIS and artificial intelligence algorithms to study the spatial and temporal evolution characteristics of air pollution in the Xiamen-Zhangzhou-Quanzhou urban agglomeration, explore the influence of different urban elements and land layouts on the spatial and temporal distribution of PM_2.5_, and, on this basis, divide the positive and negative benefit indexes of population health and cleanliness and explore the pattern of air health and safety in different regions of the Fujian Delta. This study helps people understand the potential hazards of air pollution, which is of great practical significance for taking active protective measures against air pollution and protecting the physical and mental health of the public.

## 2. Materials and Methods

### 2.1. Study Areas

The Xiamen-Zhangzhou-Quanzhou urban agglomeration, located on the southeast coast of Fujian Province (Figure 1), is one of three characteristic economic clusters rising along the coast of China. It consists of Xiamen, Quanzhou, and Zhangzhou cities and the districts and counties under their jurisdiction. The total area of this region accounts for 20% of the total area of Fujian Province. The resident population and gross regional product of the three cities in 2020 accounted for 45.74% and 48.03% of the total population in Fujian Province. The area has a high concentration of socioeconomic resources, and the economic pattern of urban agglomeration is taking shape [17].

In this paper, we selected 417 townships and street administrative units in the urban areas of the Xiamen-Zhangzhou-Quanzhou urban agglomeration to facilitate the proper alignment of subsequent planning responses with the Chinese administrative system and to coordinate the feasibility of implementing planning measures (the research object of this paper did not include Kinmen County due to the lack of data on Kinmen County).

### 2.2. Data Sources and Preprocessing

#### 2.2.1. Data Sources

The study data included PM_2.5_ site monitoring data and data on geographic elements, meteorological elements, ground cover, and socioeconomic and other natural and human elements that may have influenced the spatial heterogeneity of the data. The PM_2.5_ monitoring data were the PM_2.5_ concentration data of all ground monitoring stations in Fujian Province. We obtained GDP data from the statistical yearbook (https://tjj.fujian.gov.cn/, accessed on 3 October 2022). The data on residential areas, medical points, tourist attractions, and other points were obtained from the Baidu Map API port. Population density data were obtained from the Worldpop data platform (https://www.worldpop.org/, accessed on 5 October 2022) and refined at the gender and age levels using the six-pop data. Surface cover data were retrieved from the National Earth System Science Data Sharing Service platform, using the 30 m resolution land cover FROM-GLC dataset for 2020 [18].

#### 2.2.2. Data Sources

The preprocessing of data mainly included PM_2.5_ concentration preprocessing based on regression analysis and preprocessing elements based on the vulnerability, exposure, and adaptation of data composition.

(1)Spatial and temporal distribution of PM_2.5_ concentration pretreatment based on regression analysis

The spatial distribution of PM_2.5_ in the study area is the basis for the health pattern analysis; therefore, the spatial and temporal distribution data of PM_2.5_ in the region must first be resolved using a reasonable method. Ground monitoring stations are a direct means to obtain PM_2.5_ concentration data, but it is difficult to achieve full coverage at the regional level, because the data are limited by the spatial distribution of the monitoring stations. In order to solve this problem, academics usually start from the conversion of relevant data that may affect the spatial distribution of air pollution concentration, regression analysis simulation [19,20], and satellite remote sensing interpretation [21,22,23] in order to obtain PM_2.5_ concentration data with full coverage at the regional level. The spatial distribution of PM_2.5_ in the study area is the basis for analyzing health patterns. Therefore, it is necessary to analyze the spatial distribution data of PM_2.5_ in this area using a reasonable method.

In this paper, we invert the spatial and temporal concentrations of PM2.5 in the Fujian triangle using a land use regression model commonly used in academia [24,25]. The specific approach is: (i)34 monitoring stations in Fujian Province were selected (Figure 2), and multiply scaled multi-scale buffer zones with different radii from 250 m to 16,000 m were established with these stations as the center of the circle, and the values of meteorology and elevation, rainfall, wind speed, population, and land use type ratio in the buffer zone were counted in GIS software and used as the independent variables;(ii)Using Spss software, a model with a fit higher than 0.90 was obtained by applying stepwise regression with the PM_2.5_ concentration values of each station as the dependent variable in 2020.(iii)The model was validated by *p*-value test, covariance test, normal distribution of residuals, and D-W test of relevant parameters, and compared and validated by GEODA software.(iv)Using this model as the basis for calculation, the annual PM_2.5_ concentration spatial distribution map at the raster scale for the whole area of Xiamen-Zhangzhou-Quanzhou urban agglomeration was inverted (Figure 3).(v)On this basis, the mean values were aligned with the basic units of the townships and streets studied to eliminate the influence of the zonal area factor on the analysis results and to serve as the basis for the health pattern analysis.

(2)Data composition and preprocessing of health risks and related factors

This study focuses on the degree of benefit or impairment of physical health faced by a population in a certain air cleanliness environment due to different regional atmospheric cleanliness, which is known as the regional atmospheric health pattern. According to some scholars, it is usually constituted by a functional relationship between regional exposure, vulnerability, and adaptation [26]. Among them, it is generally believed that regional exposure mainly involves two main elements, namely pollution concentration and population density; regional vulnerability mainly refers to the density of pollution-sensitive populations or elements in a region, and in this paper, it is characterized by the spatial density of females, adolescents, and elderly people [27,28]. Regional adaptability refers to the mitigation ability of a region against pollution hazards, and in this paper, it is mainly characterized by medical services, forest land occupation ratio, and other spatial elements.

For the data above, this paper analyzed the factors affecting vulnerability, exposure and adaptability, and annual PM_2.5_ concentrations using the Spearman correlation analysis method. The results indicate that all variables are significantly correlated. Meanwhile, the variables were processed in the interval of [0, 1] to avoid the influence of different magnitudes and units among the variables on the analysis results. Descriptive statistical results and correlation analysis data are listed in Table 1.

## 3. Research Method, Evaluation System, and Technical Route

### 3.1. Methodology of Atmospheric Health Pattern Zoning

China’s Ambient Air Quality Standard (GB 3095–2012) divides ambient air functional areas into two categories: the first category is for nature reserves, scenic spots, and other areas requiring special protection, and the second category is for residential areas, mixed commercial and traffic residential areas, cultural areas, industrial areas, and rural areas. The average annual concentration limit for PM_2.5_ in the first category is 15 μg/m^3^, and 35 μg/m^3^ in the second category. According to this criterion, for a Class II area of a city, PM_2.5_ in the area is healthy and clean if the daily and annual averages are within the corresponding concentration limits. Similarly, if the population density of the region is higher, it indicates that the health benefits of a clean atmosphere in the region are higher.

We found that the annual average value of PM_2.5_ in the Xiamen-Zhangzhou-Quanzhou urban agglomeration was between 15 and 35 μg/m^3^ based on atmospheric monitoring data, although there were daily pollution concentration exceedances in the area during winter. In other words, other functional areas have already met the limits of China’s Ambient Air Quality Standards for PM_2.5_ pollution concentrations, except for nature reserves, scenic spots, and other Class I areas. In simple terms, other functional areas meet the air cleanliness level. Therefore, it is not possible to judge the impact of air pollution on public health that has been reached simply by evaluating criteria such as exposure and vulnerability beyond the corresponding pollution thresholds. Therefore, it is necessary to explore and establish a new evaluation system for air environmental quality for this situation.

Therefore, in this paper, according to the national standards mentioned above, the annual PM_2.5_ concentration limit value of 35 μg/m^3^ corresponding to the prescribed Class II area was used as the threshold value of the atmospheric health index to perform atmospheric health pattern zoning. Among these zones, an annual average concentration above this threshold was regarded as the negative interval of the atmospheric health index, i.e., the higher the population density or the longer the exposure time, the more human health damage is caused; on the contrary, the interval below this value was the positive gain interval of the atmospheric health index. Meanwhile, we divided the atmospheric health index into intervals by applying the natural breakpoint method, as shown in the following Table 2.

### 3.2. The Atmospheric Health Evaluation System

To analyze the atmospheric health pattern of the Xiamen-Zhangzhou-Quanzhou urban agglomeration, this paper refers to the specifications of the Ministry of Environmental Protection of the People’s Republic of China and constructs an atmospheric health pattern evaluation system consisting of “exposure-response intensity-regional vulnerability-regional adaptability” based on theoretical models, such as Pressure-State-Response, DPSIR, and DSR, from related papers [29] (Figure 4).

(1)Exposure–response level and classification

The exposure–response degree (ERD) measures the relationship between the spatial and temporal activity behavior of the population in the region and the environmental concentration of pollutants expressed as a function of pollutant concentration and exposure time and classified as instantaneous exposure, time-integrated exposure, and time-averaged exposure according to the exposure duration.

Studies have shown that long-term exposure to PM_2.5_, a biomarker of subclinical health hazards, is causally linked to an increase in total mortality in the region and is likely to increase the burden on the respiratory and cardiovascular systems, as well as have a more negative impact on women, children, and the elderly [30,31]. On the contrary, if there is a high level of atmospheric health for a long time, it improves the population’s overall health, reduces the risk to sensitive people, and improves the urban environmental health standard. The product of population density and pollutant concentration is usually used to express the health exposure level in a certain spatial and temporal range, but considering the current situation of cleaner air quality in the study area of this paper, combined with the individual properties and spatial distribution characteristics of PM_2.5_, and with reference to the prescribed limits of the National Standard for Ambient Air Quality of the People’s Republic of China (GB 3095–2012), this paper proposes the relationship between the annual PM_2.5_ concentration and equation of exposure-responsiveness of population density.
(1)$$erd=\left\{ \begin{array}{l} \left(35-acp\right)\times c_{i}\times dp\times c_{i},acp\leq 35\;\mu g/m^{3}\\ -\left(acp_{i}-35\right)\times c\times dp\times c_{i},acp>35\;\mu g/m^{3} \end{array}\right.$$
where *erd* is the exposure–response degree; *acp* is the standardized annual average PM_2.5_ concentration, *dp* is the standardized population density, and *c_i_* is the weight of the corresponding index.

Equation (1) uses a threshold value of 35 μg/m^3^ as a distinction, with erd greater than 0 indicating the cleanliness benefit and erd less than 0 indicating the degree of health impairment. The greater the cleanliness benefit degree index, the more the exposure–response degree tends to be positive for health promotion; the higher the absolute value of the health impairment degree, the more the exposure–response degree tends to be negative for pollution superposition. This is consistent with the previous section on health pattern partitioning, but can also ensure that the erd index converges to positive values in the clean area and negative values in the over-contaminated area to differentiate whether the area is clean or unclean.

The progressive meaning of this formula is that it avoids the errors of the previous formula for calculating contamination exposure. In the conventional formulation, since exposure is a product of pollution concentration and population density, it tends to count cities within the air cleanliness threshold, but with high population density as areas of high pollution exposure.

(2)Regional vulnerability

Regional vulnerability (RV) in this paper refers to the vulnerability of a population or socioeconomic entity to attack and damage due to pollution or risk; the greater the vulnerability, the lower the resilience of the corresponding population or socioeconomic entity. Relevant atmospheric science and epidemiological studies have shown that PM_2.5_ is likely to negatively affect the respiratory tract of susceptible populations, such as women, adolescents, and the elderly, leading to reduced lung function and cardiopulmonary system diseases [32,33]. In this paper, we selected the female population density, the population density under 14 years of age, the population density over 65 years of age, and the number of residential neighborhoods to characterize regional vulnerability.

(3)Regional adaptability

Regional adaptability (RA) emphasizes the ability of societies and populations in a region to change their state or behavior to mitigate or adapt to the health hazards of air pollution. There have been two main approaches used in research related to the mitigation of or adaptation to air pollution: one is performed at the regional or city scale to examine the amount of public resources allocated in the area regarding the enhancement of mitigation or adaptive capacities, such as the number and level of regional healthcare facilities, the scope and accessibility of urban green spaces, etc. [34,35]. The other involves considering the sensitivity, intensity, and capacity of individual responses, i.e., the individual’s ability to control the relevant resources. Therefore, this paper evaluated the regional economic level, medical facilities, and the living environment, and selects elements such as medical points, forest land share, GDP, number of tourist attractions, and research and educational institutions to reflect regional adaptability.

(4)Evaluation system weights

This paper constructed an evaluation system based on AHP cascade analysis with the exposure–response degree, regional vulnerability, and regional adaptability as the criterion layer and GDP and other variables as the indicator layer. According to the correlation analysis results above, the weight of each indicator was obtained based on expert scoring, and the consistency of the obtained judgment matrix was 0.0516, which passed the consistency test. Additionally, we calculated the results for each criterion layer according to the weight. The evaluation system is shown in the following Table 3.

Finally, based on the theoretical derivation and the index system, the atmospheric health pattern of each cell was calculated as follows:(2)$$ahp=\left\{ \begin{array}{l} erd\times c_{i}+rv\times c_{i}+ra\times c_{i},erd\geq 0\\ -erd\times c_{i}-rv\times c_{i}+ra\times c_{i},erd<0 \end{array}\right.$$
where *ahp* is the final atmospheric health pattern, *erd* is the degree of exposure–response, *rv* is the regional vulnerability, and *ra* is the regional adaptability, all of which are unitless, while *c_i_* is the weight of the corresponding indicator.

The actual meaning of the indicator in this formula is that the health pattern of an area is the sum of the three indicators of clean exposure, area vulnerability, and adaptability when the concentration of atmospheric pollution is not exceeded. It has been shown that the higher the air cleanliness, the higher the population density including vulnerable populations, and the better the health security conditions in an area, the higher the level of health patterns or health effects generated in the area. Similarly, when the concentration of atmospheric pollution exceeds the limit, the health pattern is equal to the sum of the negative values of area adaptation and area vulnerability to pollution exposure. This suggests that vulnerable populations will be the primary victims of air pollution in areas where pollution concentrations exceed the limit and that regional adaptation factors can mitigate the degree of harm.

### 3.3. Technical Routes of the Study

The technical route of the study was as follows: (1) we collected relevant data and constructed a data set and performed data processing work, such as replacing vacant values, correlation rejection, normalization, and extreme difference variation, AHP analysis, weight assignment, and calculation of weighted integrated results. (2) We analyzed the spatial and temporal PM_2.5_ concentrations in the study area. We conducted multiple linear regression using ground station data in Fujian Province to eliminate the problem of relatively few ground stations measuring PM_2.5_ in the Xiamen-Zhangzhou-Quanzhou urban agglomeration and the insufficient number of regression samples. We obtained the regression equation with the goodness-of-fit R^2^ greater than 0.90 and simulated the spatial distribution of PM_2.5_ concentrations in the study area. (3) We established an atmospheric health evaluation model, comprehensive evaluation system weight determination, evaluation of regional exposure response, vulnerability and adaptability, and analysis of the atmospheric health pattern, and made suggestions for optimization (Figure 5).

## 4. Results

### 4.1. Analysis of the Spatial Differentiation Pattern and Characteristics of Air Pollution

To investigate in depth the differences in the spatial distribution of annual mean PM_2.5_ values under the administrative units of townships in the Xiamen-Zhangzhou-Quanzhou urban agglomeration, the differences in the descriptive statistics of the administrative district of townships in the area and at the scale of three prefecture-level cities, Xiamen, Zhangzhou, and Quanzhou, were compared (Table 4).

The results show that the indicators of PM_2.5_ in the administrative districts of townships in 2020 showed some differences at different scales. The overall annual average value is 19.16 μg/m^3^, but comparing the average value within a single city, we can see that the overall average value is smaller than that for Xiamen and Zhangzhou and slightly larger than for Quanzhou. In other words, at the urban scale, Xiamen is affected by fewer statistical units, better economic development, and more PM_2.5_ generated by traffic and industrial production, which shows a slight increase in the pollution degree of the healthy air pattern. From the standard deviation dimension, the differences between township units within Xiamen are much less than those observed for other cities and on the whole, and the differences among townships in Zhangzhou are the largest. As inferred from the economic and social development of each place, Xiamen has a high level of urbanization and a developed tertiary industry, and still shows relatively high mean PM_2.5_ values and levels of homogeneity, despite its proximity to the coastline and its rich three-dimensional urban greenery. Meanwhile, Zhangzhou is vast and varies greatly, both surrounded by forest greenery and influenced by secondary industry production, showing spatial characteristics of both high and low values. The combined skewness and kurtosis demonstrate that the sample distribution in Xiamen is closest to the normal distribution; the skewness and kurtosis of each scale are concentrated around the value of 0, indicating that the sample values are close to the normal distribution, with good data uniformity, and the overall spatial quality of the region is good and relatively stable.

Combined with the PM_2.5_ annual average value classification map (Figure 6) and the two-way table of atmospheric health model zoning (Table 5) for analysis, the PM_2.5_ atmospheric cleanliness class of the Xiamen-Zhangzhou-Quan urban agglomeration in 2020 shows a spatial pattern dominated by B-1 and B-2 classes, supplemented by B-3 and A-3 classes. Of these, B-1 and B-2 were mainly concentrated in the mountainous areas of Zhangzhou and Quanzhou; the dominant zone for air cleanliness in Xiamen City is Grade B-2, which is slightly polluted. Class B-3 is mainly concentrated in Changtai County, Xiangcheng District, and Longwen District of Zhangzhou, as well as Quangang District and Hui’an County in Quanzhou.

The specific distribution takes Quanzhou City, Quan Gang District, and Huian County, Zhangzhou City, Xiang District, and Longwen District and Zhangzhou City, southern Zhaoan County, and Dongshan County as the connecting lines, dividing the city group into the spatial distribution of concentration differences between east and west. The PM_2.5_ concentration in the whole region shows a three-peak and two-valley juxtaposition. The lowest value occurs in the southwest of Pinghe County, Zhangzhou City, probably due to the influence of annual average statistics and the excellent natural environment. In contrast, the highest value is 26.11 μg/m^3^ in Wu’an Town, Changtai County, located in Guanshan Industrial Park, which has many industrial enterprises and is engaged in relatively high-pollution industries such as plastics and weaving, which in turn affects regional air quality. From an intra-city perspective, Zhangzhou City, Zhao County, Pinghe County, Xiangcheng District, and the surrounding three combined to form a triangle of high-pollution peak areas. In Quanzhou City, the concentration gradually decreased from east to west and from coastal to inland areas; Xiamen City, on the other hand, exhibits the same results as the previous one, showing light pollution in general and relative homogeneity.

To accurately grasp the spatial pattern and characteristics of air pollution partitioning, this study uses spatial autocorrelation to conduct spatial clustering analysis of annual PM_2.5_ averages and explores the interrelationships among different spatial units based on the first-order queen adjacency matrix.

The degree of global spatial autocorrelation was first analyzed, and the resulting Moran’s I index was 0.51, which showed significance at the *p*-value of 0.01 level. Since global spatial autocorrelation is difficult to clarify the features of local space, local spatial autocorrelation features were analyzed using the Local Indicators of Spatial Association (LISA) analysis map [37]. The LISA analysis map can be used to measure the relationship between a geographic unit and its surrounding areas, and verify whether these spaces are randomly distributed or clustered postures, which can provide a spatial measurement perspective to recognize the spatial distribution pattern. The analysis results obtained are shown in Figure 7.

The LISA plot (Figure 7) shows that the spatial distribution of high–high and low–low clusters shows spatial convergence. PM_2.5_, as small solid particles diffused with airflow, shows spatial dependence at the township scale and significant spatial autocorrelation characteristics. Comparing Figure 4, we can see that the high and high clustering areas show a pattern of both spatial convergence and diffusion, and generally show a tendency toward fragmented distribution along the north–south midline of the Xiamen-Zhangzhou-Quanzhou urban agglomeration. The original high-value areas in Zhangzhou City, such as Xiangcheng District, Longwen District, and Changtai County, were further examined to refine the township units with high spatial autocorrelation in the region. In contrast, the original high-value areas in the eastern part of Quanzhou City are relatively diffuse, indicating that the analytical idea of the spatial distribution of annual mean values masks the extent of the spatial diffusion influence of high-value areas on peripheral areas. Quan Gang District and Hui’an County in Quanzhou show a concentrated distribution of high-value areas. Overall, it tends to show a fragmented distribution along the north–south midline of the Xiamen-Zhangzhou-Quanzhou urban agglomeration. The low clusters are concentrated in the mountainous areas in the west, which provide a certain level of health benefit due to the relatively weak level of socioeconomic development combined with the purifying and adsorbing effects of forests and woodlands. It is worth noting that, as in the previous analysis, Xiamen City performs relatively poorly with a B-2 grade. The slightly inferior air quality in Xiamen is due to the dense population and high retention of motor vehicles, as well as the natural environment influenced by frequent winter inversions and the tendency of air pollutants to accumulate, as inferred from relevant studies.

### 4.2. Spatial Differentiation and Characterization of the Atmospheric Health Evaluation System

Based on the above evaluation system and combined with the existing data, this paper performs an analysis of the results of the atmospheric health evaluation system with the exposure–response degree, regional vulnerability, and regional adaptability as the elements.

(1) Exposure–response: Clean exposure benefits, with the region showing a “north-central-south depression, with the rest of the region [being] different and prominent”.

First, the exposure–response degree of the Xiamen-Zhangzhou-Quanzhou City cluster is analyzed. Based on the fact that the average annual PM_2.5_ value of each unit is below 35 μg/m^3^, we can infer that the region as a whole behaves as a clean area, and the classification combined with Equation (1) can be used to identify all areas as clean benefit areas and calculate the degree of cleanliness benefit.

The results are illustrated based on the quantile breakpoint method (Figure 8). The clean benefit degree of the Xiamen-Zhangzhou-Quanzhou urban agglomeration shows the characteristics of “north-central-south depression, and the rest of the region is different and prominent”, and the three major “basins” show the program of uniform distribution in the northeast, central, and west.

The high-value area generally shows an E-shaped convergence with the opening to the right; the spatial layout starts from the main island of Xiamen and is interspersed with three axes to the north, west, and south, and intersects with the mountainous areas to the west, north, and south.

Among them, the entire area of Dehua County and Anxi County in Quanzhou City shows a high degree of cleanliness benefit. The cleanliness benefit for areas outside the main island of Xiamen is also reflected in the high values. Zhangpu County, Yunxiao County, Pinghe County, Nanjing County, and Hua’an County in Zhangzhou City are strung together into a line, all showing a high level of cleanliness.

The low-value areas are mainly those with PM_2.5_ concentrations above the annual mean, heavily concentrating in Zhangzhou City’s Xiangcheng District, Longwen District, Longhai City, and Changtai County, the southern part of Zhaoan County, and the vicinity of Dongshan County, all of which show a lower level of cleanliness benefit. Some areas in Jimei District in Xiamen City show lower benefit levels; Quan Gang District and Huian County in Quanzhou City also show lower values of benefits.

The cleanliness benefit is the product of the constant of 35μg/m^3^ and the difference between the average annual PM_2.5_ and the population density, which is mainly affected by the spatial distribution characteristics of these factors. As the most densely populated areas are coastal areas or urban or rural construction land areas with dense industrial and mining enterprises, the above areas have high construction intensity, high industrial and commercial volume, and a high density of roads and traffic, causing high pollution emission levels. At the same time, the dense distribution, frequent activities, and high mobility of residents increase the possibility and length of exposure to pollutants, thus reducing the cleanliness levels of similar areas.

(2) Regional vulnerability: a trend of decay from east to west and from coastal to inland areas.

The regional vulnerability results are plotted based on the quantile breakpoint method (Figure 9). From the figure, it can be seen that the regional vulnerability shows an overall trend of decreasing from east to west and from the coast to inland.

The high-value areas are generally centered on Xiamen Island and the urban area of Xiamen City and extend horizontally inland in a finger-like manner. Among them, the Licheng District, Fengze District, Jinjiang City, Shishi City, Hui’an County, and the coastal townships of Quangang District in Quanzhou are high-value concentrated distribution areas. Xiamen City shows a “highland” trend, with the island as the “highland” and “low-value” distribution in the surrounding areas. The high-value area of Zhangzhou City shows a linear distribution pattern, with Longhai City, Longwen District, and Xiangcheng District running in a line.

The low-value areas show a spatial distribution pattern of predominantly mountainous areas in the west, with partial “infiltration” to the coast. The inland areas of the Xiamen-Zhangzhou-Quanzhou urban agglomeration, such as Dehua and Anxi counties in Quanzhou City and Hua’an and Nanjing counties in Zhangzhou City, show a low level of vulnerability. As seen from the diagram on the right side of Figure 8 and Figure 9, the combination of low total population density, a relatively reasonable age structure, and the low number of residential neighborhoods in the region in the above areas affects regional vulnerability.

Regional vulnerability is mainly related to the spatial distribution characteristics of vulnerable groups. It is inferred from the figure that high-value areas are concentrated on relatively developed coastlands. This is due to the high population density and large number of people in the above areas, and their age structure being more prone to aging, plus the abundant public service resources and supporting facilities for the elderly, causing the disadvantaged groups in the region to “vote with their feet” and further concentrate in the coastal areas. Another part of the high-value area is distributed in Dehua County and Anxi County in Quanzhou City and Yunxiao County and Dongshan County in Zhangzhou City, which are likely to be influenced by the higher aging level in the area. Therefore, in the process of implementing measures for the prevention and control of PM_2.5_ and other air pollution, we should pay more attention to the spatial and temporal distribution and migration characteristics of vulnerable people and develop targeted pollution prevention and control and health protection mechanisms according to local conditions in response to the relatively low resistance to air pollution of susceptible populations.

(3) Regional adaptability: Spatial differentiation pattern of “high in the north and low in the south, high in the east and low in the west”.

The regional adaptation results are plotted based on the quantile breakpoint method (Figure 10). Overall, the regional adaptation showed a spatial divergence pattern of “high in the north and low in the south, high in the east and low in the west,” which showed a stepwise decline from the northeast to the southwest.

The high-value areas are mainly distributed in the coastland in the north and central parts of the city, and this area extends to the north–south midline zone of the Xiamen-Zhangzhou-Quanzhou urban agglomeration. The high-value areas in Quanzhou City are mainly concentrated in other districts and counties except for the central urban areas, such as Caricheng District and Fengze District, such as Quangang District, and Hui’an County, which shows a high level of high-value distribution, while Luojiang District, Nan’an City, and Jinjiang City also show a high-value distribution. The entire area of Xiamen displays a high level of regional adaptability; the high-value areas in Zhangzhou are concentrated in Xiangcheng District, Longwen District, and the coastland of Longhai City and Dongshan County.

The low-value areas are mainly in the southern coastal and the western and northern mountainous areas. As can be seen from the figure, Dehua County, Anxi County, and other inland county-level administrative units in Quanzhou show a low level of regional adaptability. Although the green space coverage rate of the above areas is relatively high, the economic volume, scientific research, educational support, and other social and economic forces are relatively weak. The result of the comprehensive superposition is low regional adaptability. Therefore, this demonstrates that when constructing a pollution control system, we should pay attention to regions with relatively low economic development levels and consider engineering and non-engineering fields to improve the individual and even the overall adaptability level. Zhangzhou City has more low-value level areas, showing a trend from northwest to south to southeast and surrounding high-value areas.

From the components of regional adaptability, it is clear that the leading influencing factors of regional adaptability are regional economic volume, regional medical service supply capacity, research and education support capacity, and environmental friendliness. We should start from individual and social aspects, consider the integrated improvement of awareness and ability of the individual to prevent and resist pollution, enhance the endowment of the integrated layout of the public service system at the social level, and improve integration to enhance regional adaptive capacity.

### 4.3. Analysis of the Spatial Variation Pattern of Atmospheric Health Pattern and Its Causes

Based on the analysis results of the atmospheric health evaluation system, Equation (2) is used to improve the results of the statistical analysis and spatial differentiation characteristics of the atmospheric health pattern.

The analysis of Figure 11 shows that the spatial divergence of the atmospheric health pattern shows an overall decreasing trend from north to south.

Among them, the high-value area presents an “F-shaped” spatial distribution pattern.

The east–west “two axes” show a high-value spatial distribution pattern from the northwest through the west to the northeast, starting in the middle of the west and passing through the middle to the Bay Area, while the north–south “long axis” is gathered in the western front line.

In Quanzhou City, the high-value area shows a spatial pattern with a “V”-shaped distribution containing Dehua County, Anxi County, the south of Nan’an City, Jinjiang City, Hui’an County, and Quan Gang District opening to the right.

This is closely related to the high level of cleanliness benefits and regional adaptability in the above areas. Xiamen exhibits a higher level in the entire atmospheric health pattern due to its moderate cleanliness benefit, high regional adaptability, and vulnerability.

As for Zhangzhou City, the situation is relatively complex, with high-value areas relatively scattered around the municipal district and the southern region, such as Hua’an County and Nanjing County, which have a high level of the natural environment, and Yunxia County, which has a high level of regional adaptability and also partially exhibits a superior air health pattern.

On the contrary, the low-value areas show a parallel pattern of three peaks in the north, middle, and south. The low-value areas in the north are mainly in the city of LiCheng, Fengze, and Shishi in Quanzhou, which are affected by the lower level of cleanliness benefit. Although these areas perform better in regional vulnerability and adaptability, they exhibit a lower level of atmospheric health, limited by the overall environment. The low-value areas in the central part mainly include Changtai County, Longwen District, and Longhai City in Quanzhou City. Xiangcheng District experiences lower cleanliness benefits, and the rest of the above areas have weak performance due to a lack of adaptation to and mitigation of pollution. The southern low-value areas are mainly in Zhaoan and Dongshan counties, where the level of economic and social development is low, and the industrial structure is relatively solidified, thus showing a lower level of the atmospheric health pattern.

## 5. Conclusion and Discussion

### 5.1. Conclusions

(1) In this paper, a multiple linear regression model based on Land Use Regression (LUR) was selected by comparing different software to analyze the correlation between land use, elevation, meteorology, population and other factors, and PM_2.5_ concentration. A model with a goodness-of-fit exceeding 0.90 was obtained and passed the test requirements of the covariance test, *p*-value test, D-W test, etc. The validity of the model was proved by comparative validation methods, such as applying GEODA software simulation. On this basis, a regression analysis was conducted to obtain the spatial distribution data of PM_2.5_ concentrations in the study area.

(2) According to the Chinese national standard “Ambient Air Quality Standard” (GB 3095–2012), the annual average PM_2.5_ concentration threshold of 35 µg/m^3^ was used as the threshold value for whether it is clean or not, and a regional air quality evaluation system consisting of A-grade clean zone, B-grade clean zone, and exceeded pollution zone was constructed by the natural breakpoint method. The results found that the annual average PM_2.5_ values in most areas of the Xiamen-Zhangzhou-Quanzhou urban agglomeration were within the concentration limits of 15–35 µg/m^3^, and most areas were clean zones. The spatial differentiation pattern and the origin of the clean zones were also analyzed. The study shows that the air pollution pattern of each township unit has strong spatially dependent characteristics, among which there are areas of high pollution values in both Zhangzhou and Quanzhou, but in general, the overall air quality in other areas of Quanzhou and Zhangzhou is slightly better than that in Xiamen, except for areas with high pollution values.

The spatial pattern of atmospheric health in the Xiamen-Zhangzhou-Quanzhou urban cluster was analyzed based on AHP hierarchical analysis and spatial mathematical analysis to analyze the mathematical relationship between exposure, vulnerability, and adaptation in the region, and to construct a regional atmospheric health assessment system. The results show that the level of the regional atmospheric health pattern decreases from north to south, and the high-value area shows an “F”-type spatial distribution, with the “two axes” in the east–west direction from northwest to northeast and the “long axis” enclosed in the north–south direction. The “long axis” is enclosed in the western front; the low-value region shows a pattern of three peaks in the north, middle, and south.

### 5.2. Discussion

(1) As a healthy city pursuing high quality and high value, Xiamen, although already a relatively good city in terms of air quality among the high-density cities in China, still has some gaps in its air quality if we follow the WHO air quality standards; from the perspective of the Xiamen-Zhangzhou-Quanzhou urban agglomeration, there are still some areas within the zone of slight pollution, even according to the Chinese standards. Therefore, an integrated analysis should be conducted from multiple perspectives and disciplines to further improve the quality of the air environment.

(2) China’s current air quality evaluation standards are still relatively crude, so for some areas with relatively clean air quality and a limited threshold of PM_2.5_ concentration, the so-called pollution exposure cannot be calculated by simply using the product of pollution concentration and population density. We should establish the concept of healthy cleanliness and refine the evaluation system and division of healthy cleanliness. For example, we should infer the benefit rate of clean air for the population in the region based on the product of regional population density and cleanliness to provide more practical evaluation ideas and plan directions for the design of healthy cities.

## Figures and Tables

**Figure 1 ijerph-20-03340-f001:**
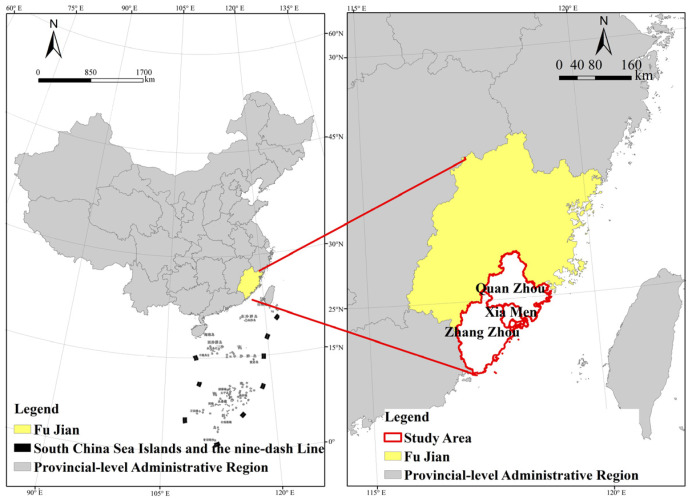
Study location map.

**Figure 2 ijerph-20-03340-f002:**
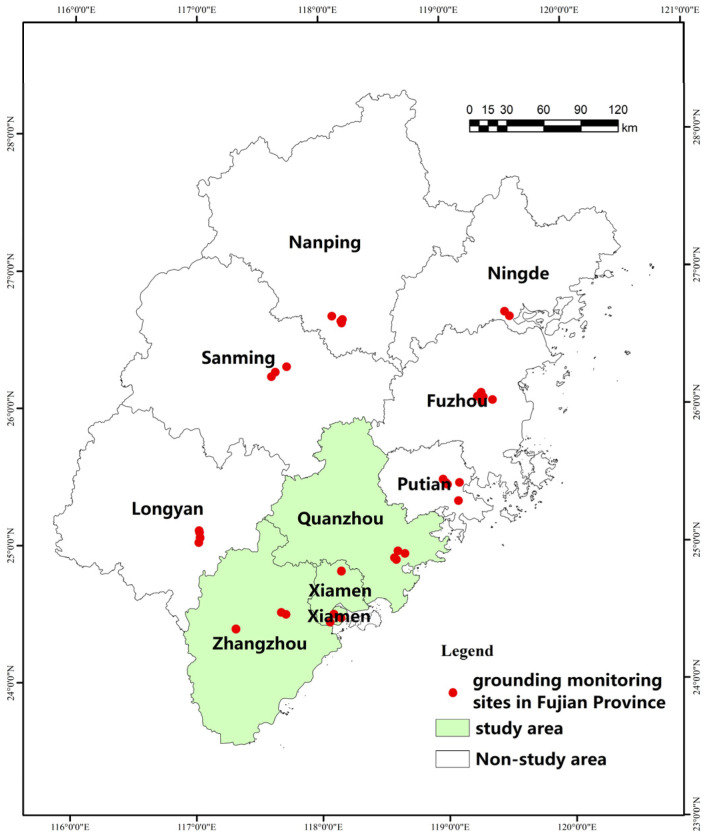
Selected atmospheric monitoring sites in Fujian Province.

**Figure 3 ijerph-20-03340-f003:**
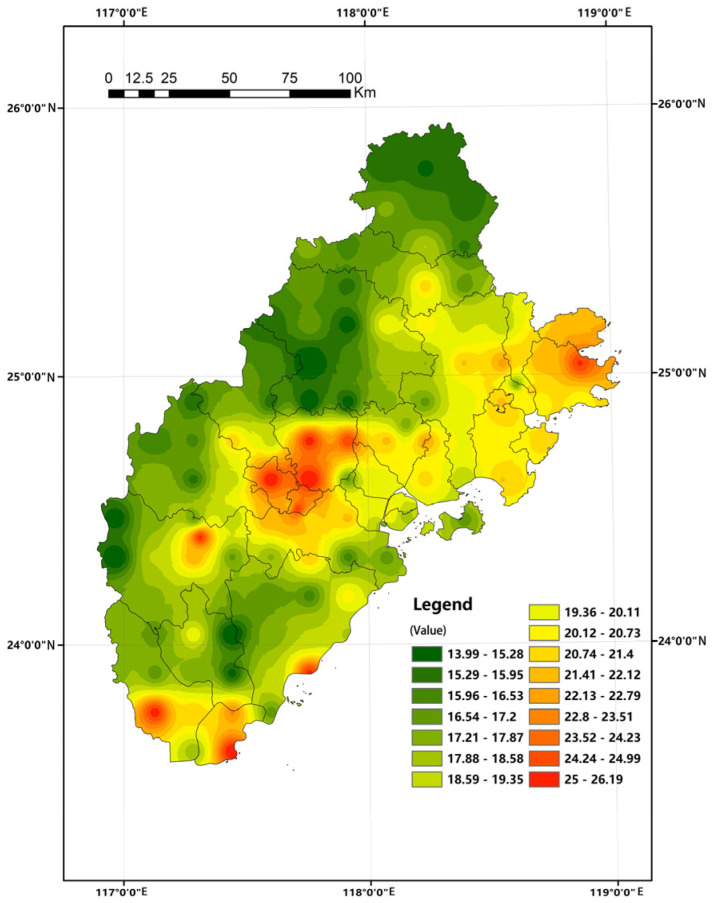
Inversion map of spatial distribution of average annual PM_2.5_ concentration in the Xiamen-Zhangzhou-Quanzhou urban agglomeration in 2020 (unit: μg/m^3^).

**Figure 4 ijerph-20-03340-f004:**
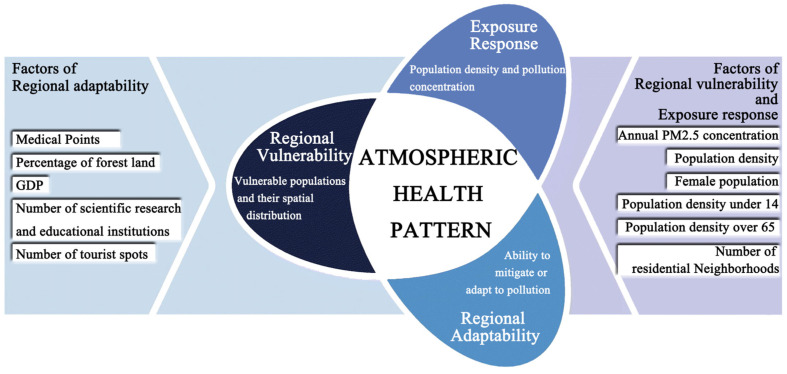
Model map of regional atmospheric health assessment.

**Figure 5 ijerph-20-03340-f005:**
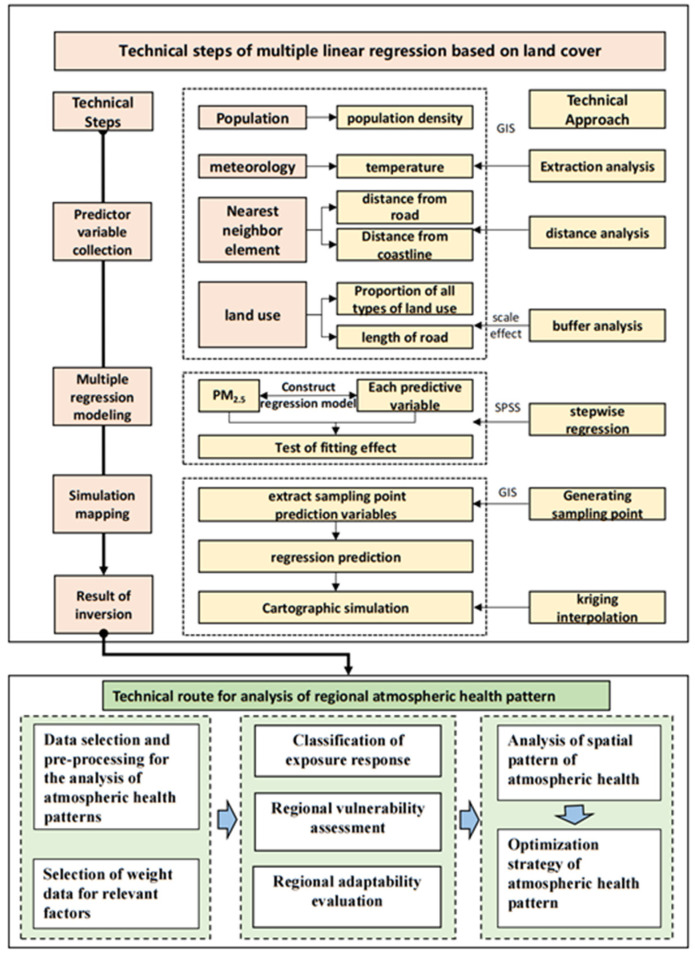
The technical routes of research. The technical flow diagram in the upper part was redrawn from [36].

**Figure 6 ijerph-20-03340-f006:**
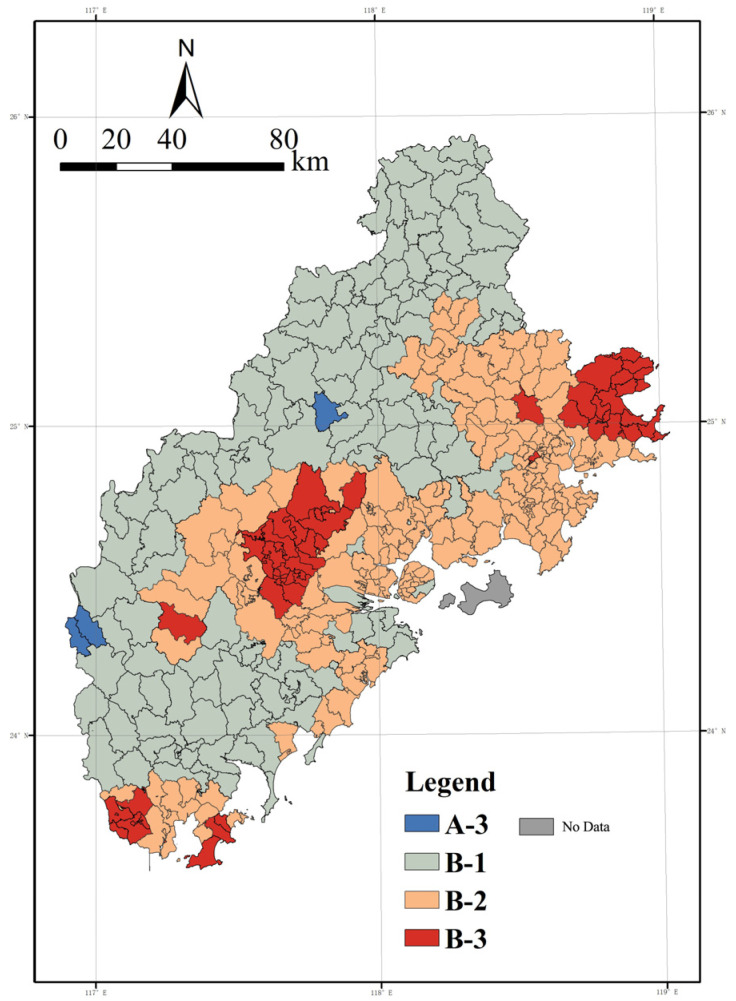
Annual average PM_2.5_ rating chart.

**Figure 7 ijerph-20-03340-f007:**
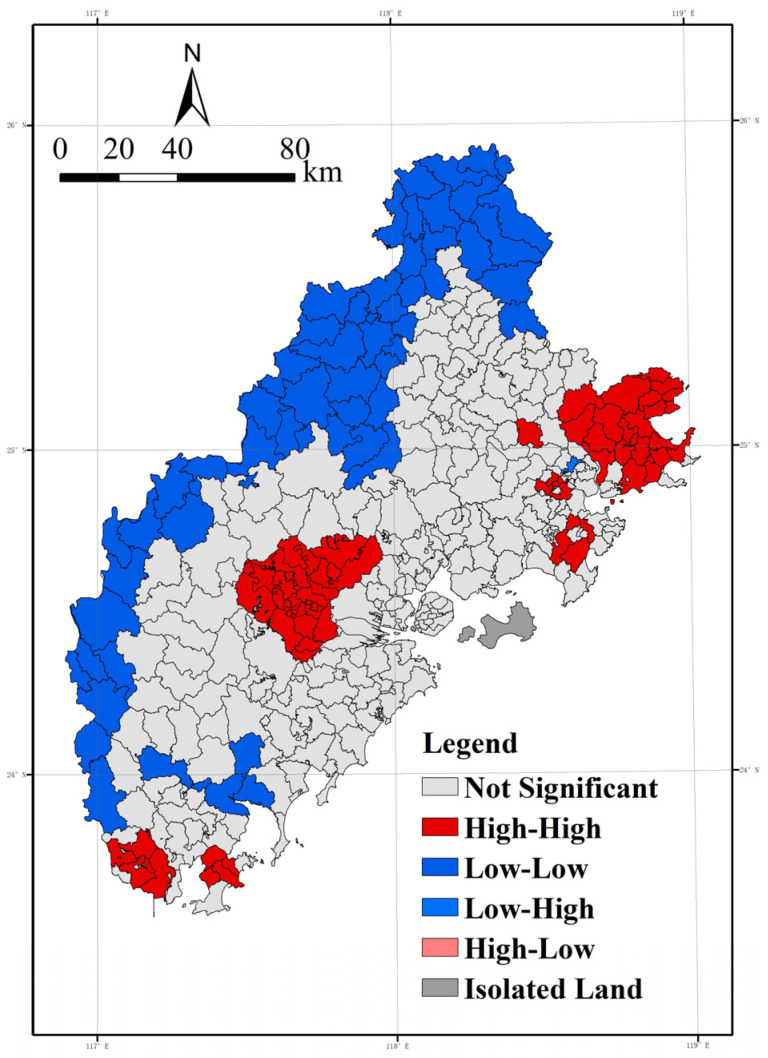
LISA of average annual PM_2.5_.

**Figure 8 ijerph-20-03340-f008:**
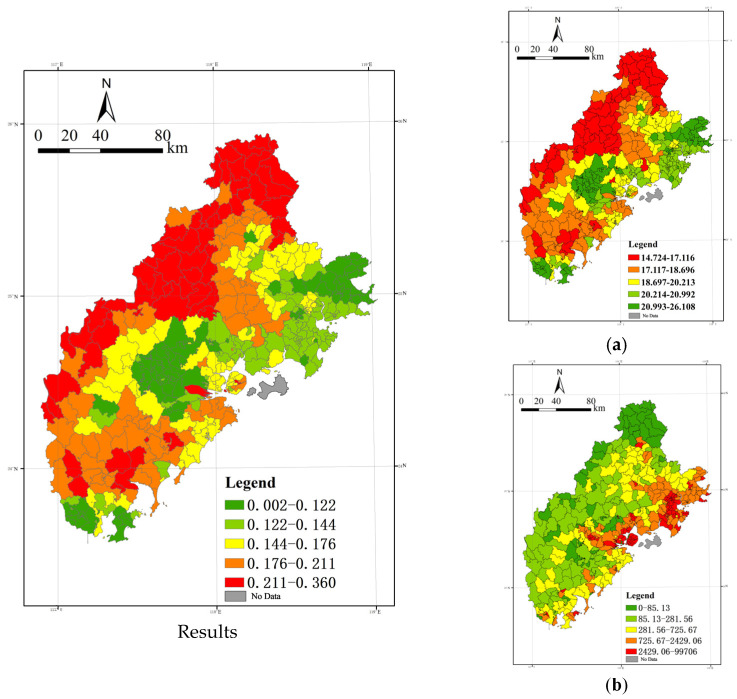
Spatial distribution of degree of benefit from cleanliness and evaluation index. (**a**) Annual PM_2.5_ concentration(unit: μg/m^3^), (**b**) Population density(Person/km^2^).

**Figure 9 ijerph-20-03340-f009:**
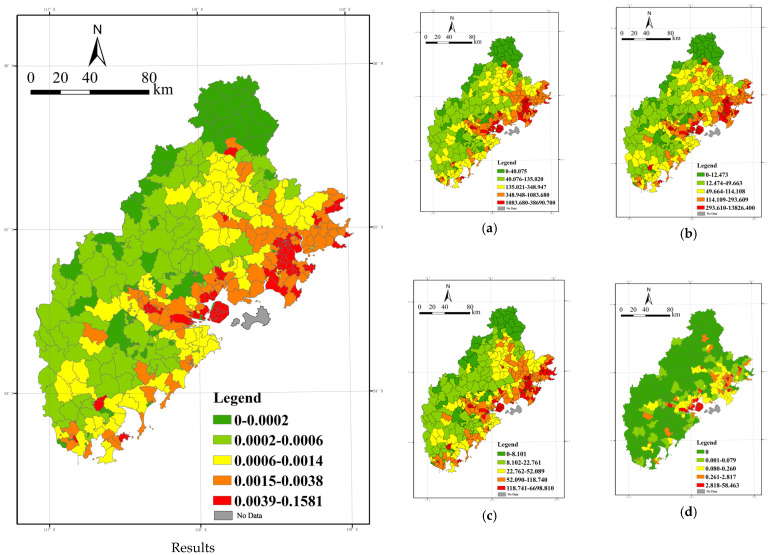
Spatial distribution of regional vulnerability and evaluation index of the female population density, the population density under 14 years of age, the population density over 65 years of age, and the number of residential neighborhoods. (**a**) Female population density (Persons/km^2^), (**b**) Population density under 14 (persons/km^2^), (**c**) Population density over 65 (persons/km^2^), (**d**) Number of residential neighborhoods.

**Figure 10 ijerph-20-03340-f010:**
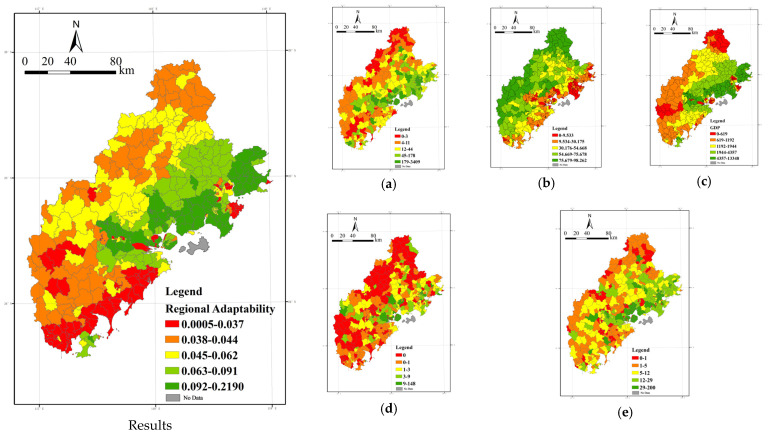
Spatial distribution of regional adaptability and evaluation index of medical points, percentage of forest land, GDP, number of tourist spots, number of scientific research publications, and educational institutions. (**a**) Medical Points, (**b**) Percentage of forest land, (**c**) GDP, (**d**) Number of tourist spots, (**e**) Number of scientific research and educational institutions.

**Figure 11 ijerph-20-03340-f011:**
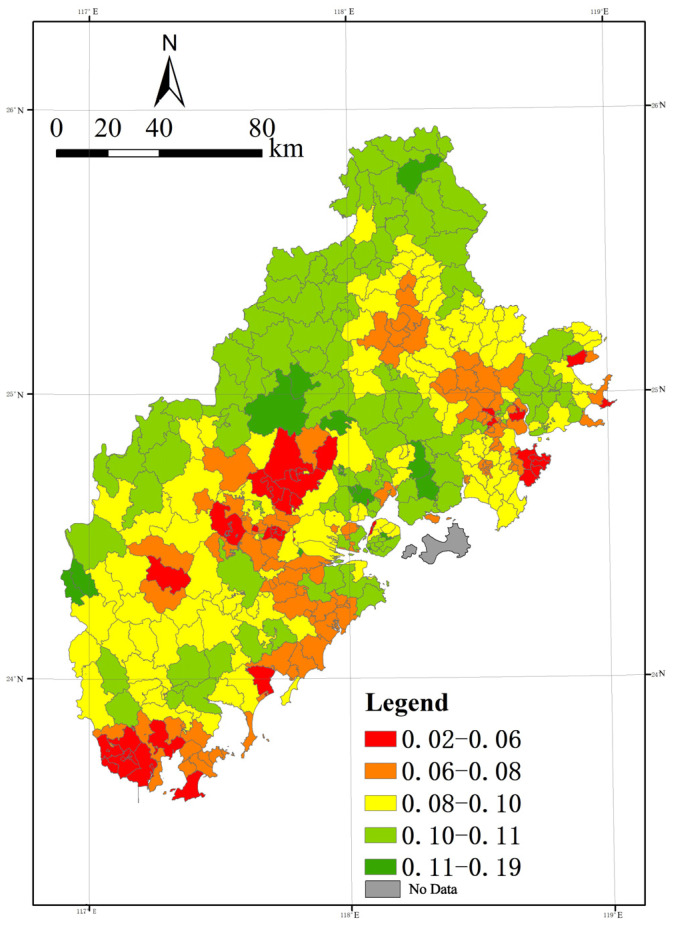
Spatial distribution of atmosphere health pattern.

**Table 1 ijerph-20-03340-t001:** Descriptive statistics and correlation test of explanatory variables.

Influence Factors	Unit	Mean	SD	Min	Max	Correlation
Population density	Person/km^2^	39,474	43,281	0	370,506	0.25 *
Female population	Person/km^2^	19,052	20,534	0	160,746	0.26 *
Population density under 14	Person/km^2^	5835	5856	0	33,468	0.21 *
Population density over 65	Person/km^2^	2538	2264	0	11,448	0.27 *
Number of residential neighborhoods	Piece	1.56	5.73	0	58	0.16 *
Medical Points	Piece	146.4	351.2	0	3409	0.22 *
Percentage of forest land	Percentage	43.41	30.34	0	98.26	−0.62 *
GDP	Ten thousand yuan	2604	2665	0	13,348	0.54 *
Number of tourist spots	Piece	4.03	13	0	148	0.07 *
Number of scientific research and educational institutions	Piece	4.03	13.25	0	148	0.20 *

Note: * indicates significance at the confidence level of 0.05.

**Table 2 ijerph-20-03340-t002:** Classification of atmospheric cleanliness patterns based on PM_2.5_.

Cleanliness Class	Zone Breakdown	Annual Average PM_2.5_ Concentration (μg/m^3^)
Class A clean area	A-1	[0, 5)
	A-2	[5, 10)
	A-3	[10, 15)
Class B clean area	B-1	[15, 18.6)
	B-2	[18.6, 21.5)
	B-3	[21.5, 35)
Pollution exceeds the standard area	C	≥35

**Table 3 ijerph-20-03340-t003:** Evaluation index system.

Guideline Layer	Weight	Indicator Layer	Unit	Weight
Exposure–response	0.4000	Annual PM_2.5_ concentration	μg/m^3^	0.2667
		Population density	Person/km^2^	0.1333
Regional vulnerability	0.2000	Female population density	Person/km^2^	0.0327
		Population density under 14	Person/km^2^	0.0556
		Population density over 65	Person/km^2^	0.0790
		Number of residential neighborhoods	Piece	0.0327
Regional adaptability	0.4000	Medical points	Piece	0.0737
		Percentage of forest land	Percentage	0.0358
		GDP	Ten thousand yuan	0.2060
		Number of tourist spots	Piece	0.0328
		Number of scientific research and educational institutions	Piece	0.0516

**Table 4 ijerph-20-03340-t004:** Descriptive statistics of town-level cities.

Group	N (pcs)	Mean (μg/m^3^)	SD (μg/m^3^)	Min (μg/m^3^)	Max (μg/m^3^)	Skewness	Kurtosis
Quanzhou	191	19.15	2.13	14.94	24.04	−0.22	1.86
Xiamen	51	19.74	0.93	17.12	21.35	−1.12	4.311
Zhangzhou	175	19.45	2.53	14.72	26.11	0.43	2.220
Total	417	19.35	2.21	14.72	26.11	0.14	2.44

**Table 5 ijerph-20-03340-t005:** Two-way table of atmospheric health pattern zoning (pcs).

Group	Low Cleanliness (A-3)	Micro Pollution (B-1)	Slight Pollution (B-2)	Mild Pollution (B-3)	Total
Xiamen	0	5	46	0	51
Quanzhou	1	75	93	22	191
Zhangzhou	2	79	56	38	175
Total	3	159	195	60	417

## Data Availability

The data presented in this study are available on request from the corresponding author.

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
