# Peer review of "Spatial Differentiation of PM2.5 Concentration and Analysis of Atmospheric Health Patterns in the Xiamen-Zhangzhou-QuanZhou Urban Agglomeration"

_ijerph, 2023, doi:10.3390/ijerph20043340_

Round 1

Reviewer 1 Report

1.     Exploring the spatial differences and health impact of PM2.5 on residents is very important to policy making to improve air quality. In my opinion, this paper can be accepted after a minor revision. The reasons are as follows.

2.     The number 2.5 in PM2.5 requires subscript. Min Delta should be defined according to Fujian Delta.

3.     Line 37-43. Both the guidelines of WHO and GB3095-2012 should be given.

4.     Figure 1 and other figures. The longitude and latitude of the map should be given.

5.     Line 140-141. How many grounding monitoring sites were used to establish regression model? The location of sites should be inserted into the map.

6.     Table 1. The unites of the influencing factors are necessary.

7.     Line 231-234. In the equation (1), there are two ci, the authors are suggested to explain. Besides, what does the c mean?

8.     Figures 4, and 6-9. The map is incomplete. Kinmen county is not included.

Author Response

Point 1: Exploring the spatial differences and health impact of PM2.5 on residents is very important to policy making to improve air quality. In my opinion, this paper can be accepted after a minor revision. The reasons are as follows.

Response 1: Thanks to the expert's recognition, we will seriously modify and strive to improve.

Point 2: The number 2.5 in PM2.5 requires subscript. Min Delta should be defined according to Fujian Delta.

Response 2: Thanks for the expert opinion. The relevant content has been modified, and the entire relevant content has been checked to ensure that it conforms to the scientific research norms and the requirements of the journal.

Point 3: Line 37-43. Both the guidelines of WHO and GB3095-2012 should be given.

Response 3: Thanks for the expert opinion. This article is supplemented by referring to the relevant contents of WHO and GB3095-2012. For a long time, the World Health Organization (WHO) has been concerned about the development of air quality standards. In 1987, WHO issued the first European air quality guidelines based on health risk assessment. On 22 September 2021, The organization's global Air Quality guidelines (AQG2021) tightened annual and 24-hour PM2.5 guidelines to 5μg/m3 and 15μg/m3, down 5 and 10μg/m3, respectively, from the 2005 version. It provides the relevant basis for the formulation of air quality standards in various countries.

According to its own development characteristics, the Chinese government updated the Ambient air quality Standard GB 3095 -- 2012 based on previous versions, which set different concentration limits for different areas. Such as the annual mean and 24-hour mean concentration limits of 15μg/m3 and 35μg/m3 in Zone 1, and the annual mean and 24-hour mean concentration limits of 35μg/m3 and 75μg/m3 in Zone 2."

Point 4: Figure 1 and other figures. The longitude and latitude of the map should be given.

Response 4: Thanks for the expert opinion. We will adjust relevant maps as required to improve the map elements.

Point 5: Line 140-141. How many grounding monitoring sites were used to establish regression model? The location of sites should be inserted into the map.

Response 5: Thanks for the expert opinion. There are 11 stations in the city cluster and 34 active stations in the province. A picture of the location of the relevant site has been formed and inserted into the article.

Point 6: Table 1. The unites of the influencing factors are necessary.

Response 6: Thanks for the expert opinion. It is true that the labeling of units was neglected here, but it has been improved in Table 3, although it has also been added in Table 1.

Point 7: Line 231-234. In the equation (1), there are two ci, the authors are suggested to explain. Besides, what does the c mean?

Response 7: Thanks for the expert opinion. ci is the weight value of the corresponding index, while c itself is meaningless and is the explanation and limitation of the elements.

Point 8: Figures 4, and 6-9. The map is incomplete. Kinmen county is not included

Response 8: Thanks for the expert opinion. The corresponding map has been updated to improve the map elements and content.

Reviewer 2 Report

The term "atmospheric health pattern" is misleading. The study only used surrogate data such as population and landuse to determine the relative risks. There is no health data involved in the study. Suggest to modify the term such as "potential exposure risk" or explain more about the term "atmospheric health pattern"

Please provide references for the statistical yearbook in Page 3 line 121 and Worldpop data platform in line 123.

Figure 2, Figure 4, Figure 6a - it should be "annual average PM25.." Please provide units

Left figure in Figures 7 and 8 needs to be clearly explained in the caption

Author Response

Point 1: The term "atmospheric health pattern" is misleading. The study only used surrogate data such as population and landuse to determine the relative risks. There is no health data involved in the study. Suggest to modify the term such as "potential exposure risk" or explain more about the term "atmospheric health pattern"

Response 1: Thanks for the expert opinion. The atmospheric health model proposed in this paper is established on the basis of the relevant theoretical basis and practical guidance, referring to the relevant provisions of WHO and China's national mandatory standards and considering that exposure of the population to a certain concentration of air pollution will cause certain damage to the body. And the analytic hierarchy process is used to logically fit and comb the "atmospheric health model". However, the abbreviation does have a certain degree of similarity with common methods in the industry, such as the analytic hierarchy process. This paper will focus on the description in the process of use and description.

Point 2: Please provide references for the statistical yearbook in Page 3 line 121 and Worldpop data platform in line 123.

Response 2: Thanks for the expert opinion. A description of the source of information has been added at the appropriate place in the article.

Point 3: Figure 2, Figure 4, Figure 6a - it should be "annual average PM25.." Please provide units

Response 3: Thanks for the expert opinion. Units have been added to the location, and the full text has been checked to fill in the gaps. In Fig. 2, Figure 4, and Fig.6a, the units are μg/m3. Figure 6 is classified according to the average annual pm2.5 concentration without units.

Point 4: Left figure in Figures 7 and 8 needs to be clearly explained in the caption

Response 4: Thanks for the expert opinion. A corresponding note has been added to the location, and the full text has been checked to add any missing points.

Reviewer 3 Report

The paper deals with a very pertinent topic. Unfortunately, the methodology is poorly described so it is very difficult to trust the described results. The presented equations seem to be based on expert opinion and not on data. Some employed methodology is not explained nor cited and it is also not explained why and how it is used. Overall the article is quite difficult to read.

- Authors say the use artificial intelligence. Where? There is no method described?

- I am not sure how and why the linear regression model is used? I understand it is used for spatial interpolation? Do you model PM2.5 concentrations at locations you do not have measurements for and do you use values of meteorology and elevation, surface cover on land use type as inputs to the model? What exactly are inputs, I guess there is a lot of them if you do step-wise selection? Step-wise selection is used to make feature selection and avoid overfitting. Do you test your model? You should have an independent test set or do some kind of cross-validation. You should first have train and test sets when you do step-wise selection and then another test set for model evaluation. I am not convinced you have a good model. R2 of 0.947 is probably evaluated on the test set.

- What are the Spearman correlations? Do you calculate Spearman correlation between the female population and PM2.5 concentrations? Why? How do you assess that the correlation is significant?

- Is Equation 1 an original contribution of this paper or is something known?

- How did you get Equation 2?

- What is in Table 4? Mean, sd of PM2.5? It is not clear.

- “To accurately grasp the spatial pattern and characteristics of air pollution partition-

ing, the study uses spatial autocorrelation to conduct spatial clustering analysis of annual

PM2.5 averages and explores the interrelationships among different spatial units based on the 1st order queen adjacency matrix.” - > authors should explain the used methodology and why they selected these methods

Minor:

- “Studies show that even short-term exposure to air pollution at concentrations exceeding the standard can cause different degrees of cough, breathing difficulties, and other symptoms.” Which studies? Which standard? Explain and reference.

- >In the Introduction section the related work is described in too much detail, but no parallel with the author’s work is given.

- It would be useful to know some facts about the investigated cities and their surrounding. What are the sizes of the cities? What is the climate and topography? What is the main industry?

- PM2.5 concentration pretreatmen? Pretreatment? You mean pre-processing?

- LISA analysis plots? Explain, sites?

- The Results section is very long and hard to read. Results could be summarized in a more concise way. 

Author Response

Point 1: The paper deals with a very pertinent topic. Unfortunately, the methodology is poorly described so it is very difficult to trust the described results. The presented equations seem to be based on expert opinion and not on data. Some employed methodology is not explained nor cited and it is also not explained why and how it is used. Overall the article is quite difficult to read.

Response 1: Thank the experts for their recognition and opinions. The formula proposed in this paper is optimized by referring to the construction idea of the analytic hierarchy process and relevant scholars' existing research on air pollution. Indeed, it neglects to explain and cite the methods used; This revision has added references to the literature and elaborated the basis of the method.

Point 2: Authors say the use artificial intelligence. Where? There is no method described?

Simulation 1

Simulation 2

Simulation 3

Simulation 4

Response 2: Thanks for the expert's opinion. The artificial intelligence method described in the preface of this paper mainly includes BP neural network simulation, which is mainly applied in part (1) of 2.2.2 (1) Pretreatment of PM2.5 concentration based on regression analysis). It is based on the annual data of several meteorological stations in Fujian Province (Fujian Triangle City Cluster in the study area of this paper is a part of Fujian Province). It tries to use BP neural network and other artificial intelligence methods and has conducted many training attempts. However, due to the limitation of the number of samples, although the goodness of fit of the obtained results is high, the randomness is strong, which is not in line with the principle of "repeatability" of scientific research work. At the same time, considering the limitation of the length of the paper, this point is not explained too much. However, in order to reflect the characteristics of "artificial intelligence" in this paper, the description of artificial intelligence methods and results is added in part (1) of 2.2.2.

Meanwhile, partial regression results of the neural network are included in this document for expert review.

Simulation 1

Simulation 2

Simulation 3

Simulation 4

As shown in the figure above, the simulation results obtained based on Matlab neural network plug-ins are quite different from each other, and it is difficult to form relatively stable simulation results in line with the principle of repeatability of scientific research. In addition, considering the stability, feasibility and popularity of the land use, a regression model was used to simulate PM2.5. Therefore, the multiple linear regression model with better goodness of fit was selected for PM2.5 prediction (i.e., pre-processing).

Point 3: I am not sure how and why the linear regression model is used? I understand it is used for spatial interpolation? Do you model PM2.5 concentrations at locations you do not have measurements for and do you use values of meteorology and elevation, surface cover on land use type as inputs to the model? What exactly are inputs, I guess there is a lot of them if you do step-wise selection? Step-wise selection is used to make feature selection and avoid overfitting. Do you test your model? You should have an independent test set or do some kind of cross-validation. You should first have train and test sets when you do step-wise selection and then another test set for model evaluation. I am not convinced you have a good model. R2 of 0.947 is probably evaluated on the test set.

Response 3: Thanks for the expert's opinion. Referring to the research of relevant papers [1], this paper conducts spatial prediction of PM2.5 based on the relatively commonly used land use regression model in the industry. This model has advantages such as convenient data acquisition, diverse input factors, and high spatial-temporal resolution simulation effect, and its accuracy and feasibility have been recognized in many studies [2]. The aim is to establish a PM2.5 concentration model in Fujian Delta .

At the same time, relevant factors such as wind speed, temperature, altitude, and the area and proportion of land use types within different buffer zones were considered, and a variety of variables were input by the stepwise regression method. Finally, the multiple linear regression model that best conforms to theoretical reasoning and has a high degree of good fit was selected as the benchmark for PM2.5 concentration prediction to avoid over-fitting or loss of important variables.

At the same time, the following multiple linear regression model is finally determined through cross-verification with the previous simulation analysis based on BP neural network (refer to the table added in this modification).

Point 4: What are the Spearman correlations? Do you calculate Spearman correlation between the female population and PM2.5 concentrations? Why? How do you assess that the correlation is significant?

Response 4: Thanks for the expert's opinion. Regression analysis is usually based on the theoretical derivation and correlation verification. In this paper, factors such as the proportion of the resident female population and the proportion of the population over 65 years old are also selected on the basis of theoretical derivation, and correlation verification is carried out, usually based on a t-test and hypothesis test is conducted under 95% confidence interval. The final results are shown in Table 1. The overall results show that the selected elements have a significant correlation with the corresponding PM2.5 concentration.

Point 5: Is Equation 1 an original contribution of this paper or is something known?

Response 5: Thank the experts for their opinion. Equation 1 is formulated according to the logical thinking of this paper, aiming to lock the interval of positive effect above 0 and limit the interval of negative effect below 0 so as to facilitate understanding. At the same time, the basic connotation of the analytic hierarchy process is referred to reducing the heterogeneity between different variables and avoiding the influence of different dimensions on the results.

Point 6: How did you get Equation 2?

Response 6: Thanks for the expert's opinion. Equation 2 is obtained according to the analytic hierarchy process and the positive and negative directions of factor action.

Point 7: What is in Table 4? Mean, sd of PM2.5? It is not clear.

Response 7: Thank the experts for their opinion. Table 4 shows the mean value of PM2.5 in the corresponding range obtained by the ArcGIS zonal statistical tool based on the township administrative unit of Mindelta Urban Agglomeration and the corresponding descriptive statistics. N is the number of samples, and the unit is each. Other indicators except kurtosis and skewness are in μg/m3.

Point 8: To accurately grasp the spatial pattern and characteristics of air pollution partitioning, the study uses spatial autocorrelation to conduct spatial clustering analysis of annual PM2.5 averages and explores the interrelationships among different spatial units based on the 1st order queen adjacency matrix.” - > authors should explain the used methodology and why they selected these methods Minor:

Response 8: Thanks for the expert's opinion. This article does neglect to introduce and explain the adoption method. The main method used by the part mentioned by the experts is the exploratory spatial data analysis method, which explores the spatial structure, spatial form, spatial trend, and outliers contained in the spatial data by using a certain degree of prior model and graphic visualization, numerical analysis, non-numerical analysis, statistics, and other methods, usually including global spatial autocorrelation and local spatial autocorrelation. Global spatial autocorrelation is mainly used to judge whether a phenomenon has agglomeration characteristics in space, while local spatial autocorrelation can reveal specific spatial correlation patterns in the study area, which are usually measured by the local Moreland index and denoted by the lisa diagram.

Point 9 :  “Studies show that even short-term exposure to air pollution at concentrations exceeding the standard can cause different degrees of cough, breathing difficulties, and other symptoms.” Which studies? Which standard? Explain and reference.

- >In the Introduction section the related work is described in too much detail, but no parallel with the author’s work is given.

Response 9: Thank the experts for their opinion. References to relevant studies were indeed omitted here and have been added in the revision.

Point 10 : It would be useful to know some facts about the investigated cities and their surrounding. What are the sizes of the cities? What is the climate and topography? What is the main industry?

Response 10: hank the experts for their input. Fujian Delta urban agglomeration is a special urban agglomeration with certain typical characteristics in China. Its particularity is manifested in the interaction of mountains, city and water, and it belongs to the subtropical monsoon climate. The region's leading industrial structure is dominated by the high-tech industry and manufacturing industries. However, this paper does lack research on related urban agglomerations of the same type or similar type, so it is supplemented in the introduction.

Point 11 : PM2.5 concentration pretreatmen? Pretreatment? You mean pre-processing?

Response 11: Thank the experts for their input. There is indeed some omission of expression here, which should be pre-processing.

Point 12 : LISA analysis plots? Explain, sites?

Response 12: LISA, short for Local Indicators of Spatial Association, is a methodology proposed by anselin in 1995; LISA actually describes the relationship between the observed value yi and the lag value yji (the values around the observed value). Spatial autocorrelation refers to the potential interdependence of observed data of some variables within the same distribution area. Tobler once pointed out that "the first law of geography: everything is related to everything else, but things near are more related than things far away." When high values are correlated with high neighborhood values or low values are correlated with low neighborhood values, the spatial autocorrelation is positive. Negative spatial autocorrelation exists when high values are correlated with low neighboring values and vice versa. Lisa is able to better represent local features.

Point 13 : The Results section is very long and hard to read. Results could be summarized in a more concise way. 

Response 13: Thanks to the experts for their advice. This article attempts to simplify the result section.

Reviewer 4 Report

Manuscript Number; IJERPH-2169972

Title; Spatial Differentiation of PM2.5 Concentration and Analysis of Atmospheric Health Pattern in Fujian Delta Urban Agglomeration

Although the topic is of interest to the Scientific community, before consideration for publication, this paper should be improved. Authors should reconsider the main objective of the paper according to the content. They should try to synthesize and emphasize the main findings of the study and avoid long sentences. Furthermore, authors should avoid drawing risky conclusions.

Evaluation; Major Revision.

1.    Firstly, the author must to follow the IJERPH guideline.

2.    Keywords; Must to revised; spelling and avoiding general and plural terms and multiple concepts (avoid, for example, 'and', 'of').

Unsuitable >>>  positive and negative gain interval;

                            Fujian Delta urban agglomeration                 

3.    Line 36 – 39; For a long time, the World Health Organization (WHO) has been concerned about instituting air quality standards. In 1987,  WHO issued the first European air quality guidelines based on health risk assessment. On  September 22, 2021, the organization issued the global Air Quality Guidelines (AQG2021).

Should be provided the level of PM25 mass concentration. e.g., World Health Organization (WHO) reports that PM2.5 levels over the WHO air quality guidelines of 5 μg·m−3.

4.    Line 56 NO2 >>> NO2 (Subscription)

5.    Table 4. Descriptive Statistics of Town-level Cities (and others)

In all of the main text, many numeric data are given with too many significant figures; 2 significant figures suffice, and 3 suffice in case the first significant figure is "1".

6.    Conclusion and Discussion need to be separate in each section.

7.    Discussion should be more detailed based on certain results or comparisons with other authors’ results.

8.    Conclusion; Many paragraphs are too short.  Please revise and combine them to only one paragraph in the conclusion. The conclusions could be further developed, there is a lot of interesting data in the article.

Author Response

Point 1: Firstly, the author must to follow the IJERPH guideline.

Response 1: Thanks for the expert opinion. We have carried out further checks on paper format etc. , according to the IJERPH guidelines.

Point 2: Keywords; Must to revised; spelling and avoiding general and plural terms and multiple concepts (avoid, for example, 'and', 'of').

Unsuitable >>>  positive and negative gain interval;

                            Fujian Delta urban agglomeration     

Response 2: Thanks for the expert opinion. We have optimized and improved the English expression and further standardized the expression of the terms you mentioned, such as positive and negative gain interval and Fujian Delta City cluster.

Point 3: Line 36 – 39; For a long time, the World Health Organization (WHO) has been concerned about instituting air quality standards. In 1987,  WHO issued the first European air quality guidelines based on health risk assessment. On  September 22, 2021, the organization issued the global Air Quality Guidelines (AQG2021).

Should be provided the level of PM25 mass concentration. e.g., World Health Organization (WHO) reports that PM2.5 levels over the WHO air quality guidelines of 5 μg·m−3.

Response 3: Thanks for the expert opinion. We have optimized this content in the original location, and the optimization results are as follows: "The World Health Organization (WHO) has long been concerned with the development of air quality standards. In 1987, the WHO first published European air quality guidelines based on health risk assessment. On 22 September 2021, The organization's global Air Quality guidelines (AQG2021) tightened annual and 24-hour PM2.5 guidelines to 5μg/m3 and 15μg/m3, down 5 and 10μg/m3, respectively, from the 2005 version. It provides the relevant basis for the formulation of air quality standards in various countries."

Point 4: Line 56 NO2 >>> NO2 (Subscription)

Response 4:Thanks for the expert opinion. We have carefully checked the full text for similar misformatting or errors to ensure compliance with scientific research specifications and journal requirements.

Point 5: Table 4. Descriptive Statistics of Town-level Cities (and others)

In all of the main text, many numeric data are given with too many significant figures; 2 significant figures suffice, and 3 suffice in case the first significant figure is "1".

Response 5:Thanks for the expert opinion. Accept expert advice, check numeric representation in the text, and lock relevant data to about 2 significant digits.

Point 6: Conclusion and Discussion need to be separate in each section.

Response 6:Thanks for the expert opinion. Adopt the expert's opinion, and modify it in the corresponding part so as to achieve a clear distinction between the conclusion and the discussion.

Point 7: Discussion should be more detailed based on certain results or comparisons with other authors’ results.

Response 7: Thanks for the expert opinion. In the result part, the comparison of similar urban agglomeration studies is added for a more detailed discussion.

Point 8: Conclusion; Many paragraphs are too short.  Please revise and combine them to only one paragraph in the conclusion. The conclusions could be further developed, there is a lot of interesting data in the article.

Response 8:Thanks for the expert opinion. This revision further extends and expands the conclusion.

Round 2

Reviewer 3 Report

I am still concerned with some aspects of this paper. Especially with the methodology used because it is not adequately described. It is still not clear how are the presented equations derived and what they represent. I still find the paper long and hard to read. Some responses are wordy but do not address my concerns. 

- So the neural network results were useless so maybe there is no need to mention them. If the values are modeled with multiple linear regression, can you describe how you trained your linear regression? How many and which inputs did you select with stepwise selection and how was R2 of 0.947 evaluated, on the train or test set? In your plots, R2 on test data is lower? How did you select the test data, since you have spatial data? Both the train and test sets are very small. This is a source of major concern.

- I do not understand this point -> "At the same time, the following multiple linear regression model is finally determined through  cross-verification with the previous simulation analysis based on BP neural network (refer to the table added in this modification)." How do you compare the results of neural networks and of MLR if you say that neural networks are not useful? What do you mean with cross verification?

- Response 4: I still did not get the answer to why you calculated the correlation between the share of the female population and PM2.5? Do women and the elderly produce more PM2.5? You say you do hypothesis testing. What is the hypothesis?  A correlation of 0.25 does not seem much of a correlation. I can understand that there should be a correlation between PM2.5 and population density, but the correlation between PM2.5 and the female population?

- Response 5. You should describe in your paper how you derive the equations.

- Response 8 is not a response to Point 8. Please explain.

- You should explain and cite in your work LISA plots.

Author Response

Point 1: I am still concerned with some aspects of this paper. Especially with the methodology used because it is not adequately described. It is still not clear how are the presented equations derived and what they represent. I still find the paper long and hard to read. Some responses are wordy but do not address my concerns. 

Response: We thank the experts for their comments. This round of revisions provides further clarification on the use of the method as well as the equations, which have been modified and explained to varying degrees in the paper.

Point 2: So the neural network results were useless so maybe there is no need to mention them. If the values are modeled with multiple linear regression, can you describe how you trained your linear regression? How many and which inputs did you select with stepwise selection and how was R2 of 0.947 evaluated, on the train or test set? In your plots, R2 on test data is lower? How did you select the test data, since you have spatial data? Both the train and test sets are very small. This is a source of major concern.

Response: We thank the experts for their comments. Regarding the issue of neural networks, since this paper did not end up with the model of neural networks for inversion, the relevant part was deleted in this revision. In addition, the other questions you raised about related issues are answered below, article by article.

  1. After comparison, this paper finally adopts SPSS software for multiple linear regression. Unlike the neural network method, it is all sample data to build the model; whether its final model is valid, i.e., the validation problem of the model, is done by various validity tests in statistics (including the goodness-of-fit test of the model, the covariance test of the data, the normal distribution test of the residuals and the D-W test, etc.).
  2. Regarding the small amount of data in the training and testing sets of BP neural network: it is limited by the distribution of sites in the study area and Fujian Province. If the BP neural network training model is used, the software will automatically put 34 sample sizes, randomly divided by 70% of the total sample size for training, 15% for validation, and 15% for prediction, which is the reason for the insufficient sample size of the validation set and prediction set, and the reason why the BP neural network method is not used in this study in the end.

However, the 34 sample size of the multiple linear regression method is sufficient to meet the industry's data volume requirements for multiple linear regression, and the number of variables is sufficient to support stepwise regression analysis because it uses the different model validation methods described above, supplemented with GEODA software tests.

  1. On the issue of choosing which elements to input step by step: In SPSS software, stepwise regression is one of the embedded functions in the multiple linear regression method, which can input all sample data at one time, and then the software will automatically exclude most of the unqualified data and build the model, but this method often cannot achieve the best results; it can also be manually matched with the stepwise regression method of the software to achieve the best fitting effect.

This study follows the method provided in the relevant references (for detailed technical procedures, see: Weng M. Li Lin. Su S-L. Case-based experimental tutorial for spatial data analysis [M]. Publication: Science Press, 2019: 185-195), the topography, meteorology, population density, the proportion of land in different radius buffers, the distance of the site from the coastline and the length of the road, etc., are categorized and gradually input into the software, and the model is built using the software and supplemented by the manual stepwise regression method.

  1. Regarding what type of elements are input: meteorological elements include average wind speed, temperature, rainfall, etc.; land types include agricultural land, forest land, grassland, water surface, bare land, and commercial, residential, industrial, transportation land, etc. (according to the classification of the Institute of Geographical Sciences and Resources of the Chinese Academy of Sciences, and similar items are appropriately combined); the proportion of land cover refers to the site as the center of the circle, different The proportion of various lands within the radius buffer zone, the radius of the buffer zone are: 250m, 500m, 1000m, etc., in order to enlarge by 2 times until 16000m;, all samples contain 34X102 elements (see the figure below).

Above: The page of the selected regression method of SPSS multiple linear regression and the part of the correlation test ticked

The above figure illustrates: the input of the independent variable factors for each sample in the initial selection in the red box, the total number of samples is 34, total 34*102 factors. As can be seen from the figure, rainfall, elevation, distance from the coastline, roads, population, wind speed, etc., are represented by the symbols in the small red box: F1, representing different land types, "F1_16000" indicates the proportion of land in a certain category within a 16000-meter radius buffer zone, and so on.

  1. Regarding the model selection of R2=0.947: Although this study provides a number of models with better fit than  R2=0.947 by stepwise regression method, SPSS, such as models 2-7, the model with more reasonable  R2=0.947 is finally selected in this paper through subsequent tests of covariance, normal distribution of residuals, etc. (as shown in the red box below).

The figure above shows the model built by stepwise regression method using SPSS software. It can be seen that the software builds models including R2 = 0.947 and higher R2.

The selected models were tested by P-value test, F-test, VIF test, etc., and finally the model satisfying all aspects and R2=0.947 was selected.

Point 3: I do not understand this point -> "At the same time, the following multiple linear regression model is finally determined through  cross-verification with the previous simulation analysis based on BP neural network (refer to the table added in this modification)." How do you compare the results of neural networks and of MLR if you say that neural networks are not useful? What do you mean with cross verification?

Response3: Thanks for the reviewer's reminder! Regarding the establishment and cross-validation of the model in this study, it is explained that â‘  on the basis of comparing the two methods of BP neural network and multiple linear regression, the method that is more applicable to the establishment of the model by multiple linear regression was selected; â‘¡ after the model was preferentially selected using SPSS, the model was further validated using the classical linear regression method of GEODA software, and reliable conclusions were drawn (see the figure below); â‘¢ using the GEODA software, the advantages and disadvantages of three methods based on classical linear regression, based on spatial lag, and spatial error were compared, and these all proved that: the R2=0.947 multiple linear regression model is reliable.

Top left: Results of validating the SPSS model based on the GEODA classical linear regression method

Top: Validation of SPSS model with GEODA spatial lag model

Upper right: results of validating the SPSS model with the GEODA spatial error model

Point 4: Response 4: I still did not get the answer to why you calculated the correlation between the share of the female population and PM2.5? Do women and the elderly produce more PM2.5? You say you do hypothesis testing. What is the hypothesis?  A correlation of 0.25 does not seem much of a correlation. I can understand that there should be a correlation between PM2.5 and population density, but the correlation between PM2.5 and the female population?

Response 4: thanks for the reviewer's reminder! â‘ It is true that the proportion of female population is not a factor affecting the distribution of PM2.5 concentration, but it is a population sensitive to air pollution such as PM2.5. In this paper, the proportion of female population and the proportion of population under 14 years old, etc., are considered as influencing factors of vulnerability in the study area, not as factors affecting the spatial and temporal distribution of PM2.5 concentrations.

â‘¡ Regarding the issue of the correlation index of female vulnerability, for the study of factors affecting health-related issues, the index of 0.26 cannot be considered too low for values in the range of variation [-1,1] during the period [-1,1], because there are so many relevant factors, which may reach several hundred. In addition, the magnitude of the value of the correlation index of a factor is closely related to the order of magnitude of that variable. In general, whether a factor should be taken into consideration or not, the usual concern should be the significance of the factor, and then the positive or negative in case of obvious significance, which is the focus of the factor should be concerned.or not, the usual concern should be the significance of the factor, and then the positive or negative in the case of obvious significance, which is the focus of the factor should be concerned.

Point 5: Response 5. You should describe in your paper how you derive the equations.

Response 5: Thanks for the expert opinion. The formula used in this paper is established based on analytic hierarchy process (AHP), referring to the idea of disaster causing factor - disaster environment - disaster bearing body in regional disaster system.

The interpretation of formula 1. Here is a calculation of the exposure response. In the industry, the product of population density and pollutant concentration is usually used to represent the exposure degree and grade of people in polluted environment. However, considering that 35μg/m3 is taken as the threshold of air cleanliness in this paper, 35μg/m3 is chosen as the basis for calculating the formula in sections.

Above the formula, that is, when acp is less than or equal to 35μg/m3, it is the clean level and the whole is positive. Therefore, (35-ACP) is used as the clean level. The larger the value is, the higher the clean level is, and then multiplied by its corresponding weight. In the lower part of the formula, when acp is greater than 35μg/m3, it is the pollution interval. In order to assign a negative value to it for the convenience of subsequent calculation, the negative number of the result of (35-acp) is taken to obtain the function expression of smaller negative value and greater pollution level, and then multiplied by the corresponding coefficient, population density and its weight. From this, formula 1 is constructed.

The actual meaning of the indicators of Equation 2 is: when the concentration of air pollution is in the absence of exceedance, the health pattern of the region is the sum of three indicators of clean exposure, regional vulnerability, and adaptability. It shows that the air pollution in a region under the limit, the higher the cleanliness of the air, including the greater population density of vulnerable people, the better the health security conditions, the higher the level of health patterns or health effects produced by the region; similarly, when the concentration of air pollution exceeds the limit, the health pattern will be impaired, it is equal to the regional adaptability and negative values of pollution exposure and negative values of the regional vulnerability sum. It shows that exposure will become detrimental in areas with excessive pollution concentrations, and the vulnerable population will become victims of air pollution, while regional adaptability mitigates its detrimental level.

Point 6: Response 8 is not a response to Point 8. Please explain.

Response 6:  Thanks for the expert's comments. Before using the spatial autocorrelation method, it is necessary to construct a spatial weight matrix between samples, and the first-order queen spatial weight matrix used here refers to one of the processing methods to study the relationship between common points and common edges between regions, such as the existence of common points or edges between two samples in space, as if they are spatially adjacent, and defined as 1 in the matrix, and vice versa as not adjacent, and defined as 0. Then based on this spatial Then, based on this spatial weight matrix, the operation of spatial autocorrelation is performed. The spatial autocorrelation method, as a common method in geography, is generally used to measure the existence of spatial dependence of the variables in a sample.

Point 7: You should explain and cite in your work LISA plots.

Response 7:  Thanks to the experts for their comments. This revision adds a citation to the explanation of the Lisa analysis method by the founder of Lisa, L Anselin (Anselin L. Local Indicators of Spatial Association-LISA [J]. Geographical Analysis, 2010,27(2):93-115.), and added some explanations of the role and significance of LISA.

In summary, to echo the requirements of the theme of this essay, this paper attempts to explore the spatial and temporal patterns of health safety in the Min Delta, with population health exposure, environmental vulnerability and adaptation to pollution as the main entry points. The technical steps of multiple linear regression were not well developed at that time in order to avoid the lack of focus of the paper. After the reviewer's reminder, the theoretical framework of health risk assessment has been added to the text, the technical steps have been explained, a more detailed and easy-to-understand technical roadmap has been redrawn, the main theories have been annotated with corresponding citations, and the conclusions at the end of the article have been refined and revised. However, due to the limitation of the length of the article, it is not possible to list a large number of technical operations in the text, and some of them can only be explained here. We would like to thank the reviewers for their careful review.

Reviewer 4 Report

This revised version is suitable for publication.

Author Response

Thank you very much for the reviewer's comments.

The reviewer's opinion is “This revised version is suitable for publication.”